# ABCG5 and ABCG8 Are Involved in Vitamin K Transport

**DOI:** 10.3390/nu15040998

**Published:** 2023-02-16

**Authors:** Michinori Matsuo, Yutaka Ogata, Yoshihide Yamanashi, Tappei Takada

**Affiliations:** 1Department of Food and Nutrition, Faculty of Home Economics, Kyoto Women’s University, Kyoto 605-8501, Japan; 2Department of Pharmacy, The University of Tokyo Hospital, Faculty of Medicine, The University of Tokyo, Tokyo 113-8655, Japan

**Keywords:** ABC protein, bile, cholesterol, transporter, vitamin K

## Abstract

ATP-binding cassette protein G5 (ABCG5)/ABCG8 heterodimer exports cholesterol from cells, while Niemann–Pick C1-like 1 (NPC1L1) imports cholesterol and vitamin K. We examined whether ABCG5/ABCG8 transports vitamin K similar to NPC1L1. Since high concentrations of vitamin K_3_ show cytotoxicity, the cytoprotective effects of ABCG5/ABCG8 were examined. BHK cells expressing ABCG5/ABCG8 were more resistant to vitamin K_3_ cytotoxicity than control cells, suggesting that ABCG5/ABCG8 transports vitamin K_3_ out of cells. The addition of vitamin K_1_ reversed the effects of ABCG5/ABCG8, suggesting that vitamin K_1_ competitively inhibits the transport of vitamin K_3_. To examine the transport of vitamin K_1_ by ABCG5/ABCG8, vitamin K_1_ levels in the medium and cells were measured. Vitamin K_1_ levels in cells expressing ABCG5/ABCG8 were lower than those in control cells, while vitamin K_1_ efflux increased in cells expressing ABCG5/ABCG8. Furthermore, the biliary vitamin K_1_ concentration in Abcg5/Abcg8-deficient mice was lower than that in wild-type mice, although serum vitamin K_1_ levels were not affected by the presence of Abcg5/Abcg8. These findings suggest that ABCG5 and ABCG8 are involved in the transport of sterols and vitamin K. ABCG5/ABCG8 and NPC1L1 might play important roles in the regulation of vitamin K absorption and excretion.

## 1. Introduction

Vitamin K is a fat-soluble vitamin involved in blood coagulation and calcification [1]. There are five types of vitamin K: vitamin K_1_, vitamin K_2_, vitamin K_3_, vitamin K_4_, and vitamin K_5_. Vitamins K_1_ (phylloquinone) and K_2_ (menaquinone) are natural vitamins, while vitamins K_3_ (menadione), K_4_ (a group of menadione esters), and K_5_ (4-amino-2-methyl-1-naphthol) are synthetic vitamins [2]. Vitamin K_1_ is mainly found in plant sources and consists of a naphthoquinone skeleton and carbon chain, while vitamin K_2_ is mainly found in animal sources and also in a Japanese traditional food, natto, which is fermented soybeans. Vitamin K functions as an activator of coagulation factors [3]. Thus, vitamin K deficiency causes coagulation defects.

ATP-binding cassette protein G5 (ABCG5) and ABCG8 are members of the ABC transporter family and are half-type ABC transporters, which consist of six transmembrane helices and one nucleotide-binding domain. ABCG5 and ABCG8 form a heterodimer (ABCG5/ABCG8) [4,5]. Mutations in either ABCG5 or ABCG8 cause a genetic disorder, sitosterolemia. Patients with sitosterolemia show high plant sterol levels and develop hypercholesterolemia and atherosclerosis at a young age [6]. Serum levels of plant sterols, including sitosterol and campesterol, are generally low. However, patients with sitosterolemia cannot eliminate plant sterols from their bodies. This is because ABCG5/ABCG8 transports plant sterols out of cells and mutations in the protein impair the transport function. ABCG5/ABCG8 is expressed in the intestine and liver of human [7]. ABCG5/ABCG8 transports not only plant sterols but also cholesterol [8]. Thus, ABCG5/ABCG8 transports these sterols out of enterocytes and suppresses the absorption of sterols in the intestine. ABCG5/ABCG8 transports sterols from hepatocytes to the bile duct to remove sterols from the body.

Niemann–Pick C1-like 1 (NPC1L1) is involved in the absorption of cholesterol and plant sterols [9,10,11]. NPC1L1 is a transmembrane protein that is homologous to NPC1, which functions in cholesterol export from the lysosomes. NPC1L1 was originally identified as a target protein of ezetimibe, an inhibitor of cholesterol absorption. NPC1L1 is also involved in cholesterol reabsorption in the liver. NPC1L1 imports sterols and ABCG5/ABCG8 exports sterols in enterocytes and hepatocytes, suggesting that sterol levels are regulated by fine-tuning of NPC1L1 and ABCG5/ABCG8. In addition to sterols, NPC1L1 absorbs vitamin K in the intestine [12]. Since substrate specificities of NPC1L1 and ABCG5/ABCG8 are similar, we speculated that ABCG5/ABCG8 also transports vitamin K. Recently, ABCG5/ABCG8 has been reported to transport vitamin D in the intestine [13]. This supports the idea that ABCG5/ABCG8 transports fat-soluble vitamins, such as vitamin K.

In this study, we examined whether ABCG5/ABCG8 is involved in vitamin K transport and showed that overexpression of ABCG5/ABCG8 repressed the cytotoxicity of vitamin K_3_ and decreased intracellular vitamin K_1_ levels. This research will be useful to understand vitamin K kinetics in the body.

## 2. Materials and Methods

### 2.1. Materials

Mouse anti-ABCA1, rabbit anti-ABCG4, and mouse anti-ABCG5 antibodies were prepared, as described previously [4,8,14]. Rabbit anti-ABCG1 (sc-20795), anti-ABCG8 (NB400-110), and mouse anti-vinculin (V9131) antibodies were purchased from Santa Cruz Biotechnology (Santa Cruz, CA, USA), Novus Biologicals (Littleton, CO, USA), and Sigma-Aldrich (St. Louis, MO, USA), respectively. Other chemicals were purchased from Sigma-Aldrich, GE Healthcare (Little Chalfont, UK), Cayman Chemical (Ann Arbor, MI, USA), Wako Pure Chemical Industries (Osaka, Japan), and Nacalai Tesque (Kyoto, Japan).

### 2.2. Cell Culture

Baby hamster kidney (BHK) cells were cultured in Dulbecco’s modified Eagle’s medium (DMEM) supplemented with 10% (*v*/*v*) fetal bovine serum (FBS) in 5% CO_2_ at 37 °C.

### 2.3. Plasmids and Transfection

Cells were transfected with expression vectors for ABCG5 (pcDNA3.1/ABCG5) and ABCG8 (pcDNA3.1/ABCG8), as described previously [4,8]. Stable cell lines expressing ABCG5 and ABCG8 (BHK/ABCG5 + ABCG8) were selected by incubating cells with 5 μg/mL blasticidin and 1 mg/mL zeocin.

### 2.4. Cellular Lipid Release Assay

Cells were incubated in the presence of 0.2% bovine serum albumin (BSA) plus 6 mM sodium taurocholate for 4 h in DMEM. The cholesterol content of the medium was determined using a fluorescence enzyme assay [15].

### 2.5. Immunoblotting

Cells were washed with phosphate-buffered saline (PBS) and lysed in lysis buffer (50 mM Tris-Cl (pH 7.5), 150 mM NaCl, and 1% Triton X-100) containing 100 μg/mL 4-amidinophenylmethanesulfonyl fluoride, 2 μg/mL leupeptin, and 2 μg/mL aprotinin. Samples were electrophoresed on 10% sodium dodecyl sulfate-polyacrylamide gels and detected using anti-ABCA1 (1:5000 dilution), ABCG1 (1:1000 dilution), ABCG4 (1:1000 dilution), ABCG5 (1:3000 dilution), ABCG8 (1:1000 dilution), or vinculin (1:20,000 dilution) antibodies.

### 2.6. The Cytotoxicity Assay

BHK cells were sub-cultured in 96-well plates at a density of 5.0 × 10^3^ cells. After incubation for 24 h, BHK cells were incubated for 16 h in DMEM containing 0.02% BSA and 10 nM mifepristone to induce ABC proteins. The cells were incubated in DMEM containing 10% FBS in the presence of the indicated concentrations of vitamin K_1_ or K_3_ for 24 h. MTT was added to the medium at a concentration of 0.5 mg/mL and further incubated for 2.5 h. Cells were washed with PBS, formazone was solubilized with dimethyl sulfoxide (DMSO), and absorbance was measured at 535 nm.

### 2.7. Transport Assay

BHK cells were sub-cultured in 6-well plates at a density of 2.0 × 10^4^ cells. After incubation for 24 h, cells were incubated for 16 h in DMEM containing 0.02% BSA and 10 nM mifepristone to induce ABCG5/ABCG8. The cells were incubated in DMEM containing 10% FBS in the presence of 100 μM vitamin K_1_ for 24 h. The cells were then washed twice with DMEM containing 10% FBS and incubated with 1 mL of DMEM containing 10% FBS for 4 h. Vitamin K_1_ was extracted from the conditioned medium and cells using methanol/chloroform. The vitamin K_1_ content in the medium and cells was measured using high-performance liquid chromatography (HPLC) analysis using a GL-7400 (GL Sciences, Tokyo, Japan).

### 2.8. Quantification of Vitamin K_1_ Using HPLC

Samples extracted using the organic solvent described above were dissolved in 500 μL methanol–chloroform (1:2, *v*/*v*). Then, the samples were applied to a COSMOSIL 5C18-MS-II packed column (4.6 mm I.D. × 150 mm) (Nacalai Tesque, Kyoto, Japan), which was prewashed with methanol–ethanol (3:1, *v*/*v*). Samples (100 μL each) were applied to the column and eluted using methanol–ethanol (3:1, *v*/*v*). The peaks were detected using UV monitor (280 nm) for quantification of vitamin K_1_ by HPLC.

### 2.9. Experimental Animals

Abcg5/Abcg8-deficient mice (B6; 129S6-Abcg5/Abcg8tm1 Hobb/J) were purchased from Jackson Laboratory (Bar Harbor, ME, USA) and backcrossed with wild-type C57BL/6 J mice (Japan SLC, Inc., Shizuoka, Japan) before use. Mice were housed in temperature- and humidity-controlled animal cages with a 12 h dark/light cycle and had free access to water and animal chow (FR-1, Funabashi-Farm, Chiba, Japan).

### 2.10. Vitamin K_1_ Administration and Sample Collection

Fifteen- to eighteen-week-old male wild-type and Abcg5/Abcg8-deficient mice were anesthetized by isoflurane and administered 25 mg/g body weight of vitamin K_1_ via intravenous injection (Nippon Zenyaku Kogyo, Fukushima, Japan). After the vitamin K_1_ administration, mice were maintained under anesthesia throughout. One hour after administration, the cystic duct was ligated and a common bile duct fistula was created using a Teflon catheter (UT-03; Unique Medical Co, Ltd., Tokyo, Japan) to collect hepatic bile specimens for 1.5 h. After bile collection, the mice were sacrificed by whole blood sampling followed by liver isolation. All specimens were stored at −80 °C until analysis.

All animal experiments were conducted in accordance with the US National Institutes of Health Guide for the Care and Use of Laboratory Animals and with protocols approved by the Animal Studies Committee of the University of Tokyo (P17-063).

### 2.11. Sample Preparation for Quantification of Vitamin K_1_

Serum specimens were diluted 100-fold with Milli-Q water. Bile or diluted serum samples (50 μL each) were transferred into a brown-colored glass tube. Subsequently, 450 µL of Milli-Q water, 500 μL of ethanol, and 15 μL of isopropanol containing 50 µM menaquinone-7 (MK-7) as the internal standard was added. After brief mixing, 2 mL of n-hexane was added and mixed using a vortex mixer for 10 min. The mixed solution was centrifuged at 2000 rpm for 5 min at 4 °C. The supernatant was transferred to a new glass tube, and the bottom layer was re-extracted with another 2 mL of n-hexane. The mixing and centrifugation conditions were the same as those used for the first extraction. The collected n-hexane layer was then dried under nitrogen gas at 30 °C. The residue was reconstituted in 75 µL isopropanol–ethyl acetate (4:1, *v*/*v*) for the quantification of vitamin K_1_ using an ultra-performance liquid chromatography (UPLC) system, as described below.

Liver specimens (0.25 g) were pulverized thoroughly with 2 g anhydrous sodium sulfate and transferred into a brown-colored glass tube. The homogenate was added to 75 µL of isopropanol containing 50 μM MK-7 as the internal standard, with 0.25 mL of ethanol and 2.25 mL of acetone, then mixed using a vortex mixer for 15 min, and centrifuged at 2000 rpm for 5 min at 4 °C. The upper layer was transferred to a new brown-colored glass tube, and the lower layer was mixed again with 0.25 mL of ethanol and 2.25 mL of acetone for the second extraction. The mixing and centrifugation conditions were the same as those used for the first extraction. The upper layers of the first and second extractions were mixed thoroughly and dried with nitrogen gas at 30 °C. The residue was dissolved in 1 mL of Milli-Q water and 4 mL of n-hexane, mixed using a vortex mixer for 10 min, and centrifuged at 2000 rpm for 5 min at 4 °C. The upper layer was transferred to a new glass tube, and the lower layer was extracted again using 4 mL of n-hexane. The collected n-hexane layer was dried using nitrogen gas at 30 °C until its volume was decreased to approximately 3 mL. The sample was then applied to a Sep-Pak^®^ silica cartridge (200 mg/3 mL) (Waters, MA, USA) connected to a Waters Extraction Manifold (Waters), which was washed prior with 3 mL of diethyl ether:n-hexane (1:1, *v*/*v*), and then 3 × 3 mL n-hexane. After sample application, the cartridge was washed with 3 × 3 mL n-hexane. Then the sample was eluted with 3 mL diethyl ether:n-hexane (3:97, *v*/*v*). The eluate was then dried under nitrogen gas at 30 °C. The residue was reconstituted in 75 µL of isopropanol–ethyl acetate (4:1, *v*/*v*), and then diluted 100-fold with isopropanol–ethyl acetate (4:1, *v*/*v*) for the quantification of vitamin K_1_ using the UPLC system, as described below.

### 2.12. Quantification of Vitamin K_1_ Using the UPLC System

The UPLC system consisted of the ACQUITY UPLC sample manager and a binary solvent manager (Waters). Sample separation was performed using a VanGuard BEH C18 (1.7 µm) as the precolumn (Waters) and an ACQUITY UPLC BEH C18 (1.7 µm, 2.1 mm × 100 mm) column (Waters) as the main column. Vitamin K_1_ and MK-7 were detected with the ACQUITY UPLC fluorescent detector (Waters) after post-column reduction using CQ-R 2.0 × 20 mm column (OSAKA SODA, Osaka, Japan). The column temperature was maintained at 50 °C during analysis. The mobile phase was a mixture of Milli-Q water (solvent A) and liquid chromatography-grade methanol (solvent B). The extraction and emission wavelengths of the fluorescence detector were set to 244 nm and 430 nm, respectively.

### 2.13. Statistical Analysis

Values are presented as mean ± SD. Statistical significance among groups was determined using ANOVA followed by Dunnett’s test or Tukey’s test. Statistical significance was set at *p* < 0.05.

## 3. Results

### 3.1. Vitamin K_3_ Is Transported by ABCG5/ABCG8

To examine if ABCG5 and ABCG8 were involved in vitamin K transport, we established cell lines expressing functional ABCG5/ABCG8. Because we and others have successfully investigated cholesterol transport using BHK cells, in which ABC transporters can be induced by mifepristone, we established BHK cell lines in which ABCG5/ABCG8 is inducible. BHK/ABCG5 + ABCG8 cells expressed both ABCG5 and ABCG8, as detected by Western blotting (Appendix A). As reported previously, BHK/ABCA1, BHK/ABCG1, and BHK/ABCG4 cells mediated cholesterol efflux (Appendix A). We reported that apolipoprotein A-I functions as a cholesterol acceptor for ABCA1 and high-density lipoprotein (HDL) functions as a cholesterol acceptor for ABCG1 and ABCG4, respectively [14]. As cholesterol efflux can be investigated using taurocholate [16], we used taurocholate as a cholesterol acceptor to compare the cholesterol efflux by ABCG5/ABCG8 and other ABC transporters. BHK/ABCG5 + ABCG8 cells also mediated cholesterol efflux to taurocholate (Appendix A), suggesting that ABCG5/ABCG8 expressed in BHK cells is functional. Next, we explored the resistance of cells expressing ABC transporters to vitamin K_3_ cytotoxicity. Vitamin K_3_ is a synthetic vitamin that is cytotoxic to cells at high concentrations [1,17]. We compared ABCG5/ABCG8 with other cholesterol transporters, ABCA1, ABCG1, and ABCG4, to examine whether cholesterol transport might affect vitamin K movement, because cholesterol affects cell membrane integrity and fluidity. When vitamin K_3_ was added to BHK/mock cells, cytotoxicity was observed at concentrations above 7.5 μM, as determined using the MTT assay (Figure 1). Vitamin K_3_ showed similar cytotoxicity for BHK cells expressing ABCA1, ABCG1, or ABCG4. Because ABCA1, ABCG1, and ABCG4 are cholesterol transporters, cholesterol transport itself does not affect the cytotoxicity of vitamin K_3_. However, BHK cells expressing ABCG5/ABCG8 were resistant to 7.5 μM vitamin K_3_. The viability of BHK/ABCG5 + ABCG8 cells was also higher than that of BHK/mock cells and BHK cells expressing ABCA1, ABCG1, or ABCG4. This suggests that vitamin K_3_ concentrations in cells expressing ABCG5/ABCG8 are lower than those in cells expressing other ABC transporters, because ABCG5/ABCG8 transports vitamin K_3_ from the cytosol to the medium. To examine the possibility that ABCG5/ABCG8 transports vitamin K_1_, we measured the cytotoxicity of vitamin K_3_ in the presence of excess vitamin K_1_ (Figure 2), because vitamin K_1_ did not produce cytotoxic effects in BHK cells up to 100 μM. BHK/mock and BHK/ABCG5 + ABCG8 cells showed a similar susceptibility to vitamin K_3_ in the presence of vitamin K_1_. This result suggests that ABCG5/ABCG8 transports excess vitamin K_1_ rather than vitamin K_3_.

### 3.2. Vitamin K_1_ Is Transported by ABCG5/ABCG8

To examine vitamin K_1_ efflux by ABCG5/ABCG8, we conducted an in vitro vitamin K_1_ transport assay. Vitamin K_1_ was added to the cell culture media, and BHK cells were incubated in the presence of vitamin K_1_ for 24 h; then vitamin K_1_ in the medium was removed by washing. After washing, cells were incubated in the cell culture media for 4 h and the vitamin K_1_ content in the medium and cells was measured by HPLC after extraction of vitamin K_1_ with organic solvents. The accumulated vitamin K_1_ content in BHK/ABCG5 + ABCG8 cells was lower than that in BHK/mock cells (Figure 3A). In contrast, the efflux of vitamin K_1_ from BHK/ABCG5 + ABCG8 cells was higher than that from the BHK/mock cells (Figure 3B). These results suggest that vitamin K_1_ is exported from the cells by ABCG5/ABCG8.

### 3.3. Biliary Excretion of Vitamin K_1_ Is Lower in Abcg5/Abcg8 -Deficient Mice

ABCG5/ABCG8 is suggested to be involved in the transport of vitamin K in vitro, but it is not clear if this is the case in vivo. We investigated the distribution and content of intravenously administered vitamin K_1_ in Abcg5/Abcg8-deficient mice and compared them to those in wild-type mice. Biliary vitamin K_1_ concentrations in Abcg5/Abcg8-deficient mice significantly decreased, and vitamin K_1_ concentrations in the liver decreased, while those in serum did not differ between Abcg5/Abcg8-deficient and wild-type mice (Figure 4A). The differential concentrations of vitamin K_1_ in the bile between Abcg5/Abcg8-deficient and wild-type mice may be partly explained by decreased vitamin K_1_ concentrations in the livers of Abcg5/Abcg8-deficient mice. However, the decrease in biliary vitamin K_1_ concentrations was more remarkable than the decrease in hepatic vitamin K_1_ concentrations. Furthermore, the total amount of excreted vitamin K_1_ during bile collection (1.5 h) in Abcg5/Abcg8-deficient mice appeared relatively reduced, while the total amounts of vitamin K_1_ in the whole liver and serum did not differ between Abcg5/Abcg8-deficient and wild-type mice (Figure 4B). These results suggest that the absence of Abcg5/Abcg8 reduces biliary concentrations (biliary excretion amounts) of vitamin K_1_, and that Abcg5/Abcg8 is involved in the transport of vitamin K_1_ from hepatocytes to the bile duct.

## 4. Discussion

In this study, we showed that ABCG5/ABCG8 was involved in the transport of vitamin K in vitro and in vivo. We have demonstrated that cells expressing ABCG5/ABCG8 are resistant to vitamin K_3_ toxicity (Figure 1) and accumulate less vitamin K_1_ in cells (Figure 3). We also demonstrated that Abcg5/Abcg8-deficient mice had less vitamin K_1_ in the bile (Figure 4). These results suggest that ABCG5/ABCG8 transports vitamin K_1_ and K_3_ out of the cells.

ABCG5 and ABCG8 transport cholesterol out of cells in the intestine and liver, whereas NPC1L1 is involved in the absorption of cholesterol. Thus, cholesterol absorption and excretion in the body are regulated by NPC1L1 and ABCG5/ABCG8. NPC1L1 is involved in the absorption of cholesterol and vitamin K. In this study, we showed that ABCG5/ABCG8 is involved in vitamin K efflux. Therefore, the absorption and excretion of vitamin K in the body is regulated by NPC1L1 and ABCG5/ABCG8 (Figure 5). This finding is physiologically relevant. The absorption of vitamin K seems to be regulated only by the importer, NPC1L1. However, it is essential that both the importer and exporter function appropriately in vitamin absorption. When vitamin K levels are low, NPC1L1 may be induced and vitamin K is absorbed. Absorption of vitamin K cannot be stopped shortly after sufficient vitamin K is absorbed, and exporters promptly remove the excess vitamin K. Scavenger receptor class B type I (SR-BI), expressed on the apical membrane of enterocytes and hepatocytes, is involved in cholesterol transport. SR-BI reportedly takes up vitamin K_1_ [18]. Since cholesterol transport by SR-BI is bidirectional, SR-BI-mediated transport of vitamin K_1_ may be bidirectional. Thus, SR-BI may also be involved in the absorption and excretion of vitamin K_1_, although NPC1L1 may play a major role in vitamin K_1_ absorption.

We have shown that BHK cells expressing ABCG5 and ABCG8 are resistant to vitamin K_3_ cytotoxicity (Figure 1). This suggests that ABCG5 and ABCG8 transport vitamin K_3_ out of the cells. Since vitamin K_3_ functions as an anti-tumor agent and cisplatin—an anticancer drug—resistant cells were also resistant to vitamin K_3_ treatment [19], ABCG5/ABCG8 may play a role in resistance of cancer cells to vitamin K_3_ cytotoxicity. ABCG5/ABCG8 seems to transport vitamin K_3_, but this is indirect evidence. We did not determine the amount of vitamin K_3_ using HPLC analysis due to the limit of detection. The transport of vitamin K_3_ should be analyzed in future studies.

We have not examined the transport of vitamin K_2_ in this study. Judged from the similarity of molecular structures of vitamins K_1_ and K_3_, vitamin K_2_ may also be a transport substrate of ABCG5/ABCG8. Vitamin K_2_ plays a role in cardiovascular health by regulating calcium homeostasis [20]. Vitamin K_2_ regulates calcification by activating matrix Gla protein, which prevents vascular calcification, and growth arrest-specific 6, which affects vascular smooth muscle cell apoptosis and movement [21]. In the case that ABCG5/ABCG8 transports vitamin K_2_, defects in vitamin K_2_ transport by ABCG5/ABCG8 may be associated with the progression of atherosclerosis in addition to cholesterol transport impairment.

We showed that ABCG5/ABCG8-overexpressing cells exported more vitamin K_1_ than control cells (Figure 3). Vitamin K_1_ is a hydrophobic compound and can pass through the plasma membrane via passive diffusion. Furthermore, vitamin K_1_ is expected to attach to plasma membranes after washing. Therefore, it is difficult to estimate the actual amounts of vitamin K_1_ transported by ABCG5/ABCG8. However, the fact that vitamin K_1_ content in the medium increased and those in cells decreased in overexpressing cells suggests that ABCG5/ABCG8 transports vitamin K_1_ using energy from ATP hydrolysis and removes the vitamin from cells. The limitation of this study is that we used BHK cells instead of intestinal cells, hepatocytes, or cell models for the intestine and liver. The use of a transepithelial transport system will be beneficial to further understand the role of ABCG5/ABCG8 in the intestine and liver.

Vitamin K_1_ concentration in the bile of Abcg5/Abcg8-deficient mice was lower than that in wild-type mice (Figure 4A), suggesting that vitamin K_1_ efflux from hepatocytes to the bile duct was impaired and that Abcg5/Abcg8 is involved in the transport of vitamin K_1_ in the liver. Furthermore, biliary vitamin K_1_ levels were lower than those in wild-type mice (Figure 4B). This also suggests the involvement of Abcg5/Abcg8 in vitamin K_1_ kinetics in the liver and bile ducts. It is possible that vitamin K_1_ excretion to the bile is impaired in patients with sitosterolemia. Intestinal and hepatic ABCG5/ABCG8 transport cholesterol and plant sterols, suggesting that intestinal and hepatic ABCG5/ABCG8 have the same substrate specificity. Therefore, we believe that intestinal ABCG5/ABCG8 is also involved in vitamin K_1_ transport in vivo.

NPC1L1 imports not only cholesterol and vitamin K, but also vitamin E. Two ABC transporters, ABCA1 and ABCG1, export vitamin E in addition to cholesterol [22,23]. This raises the possibility that ABCG5/ABCG8 also exports vitamin E, in addition to sterols, vitamin D, and vitamin K. If so, NPC1L1 and ABCG5/ABCG8 may regulate the absorption of fat-soluble vitamins, including vitamins K, D, and E. Further work is needed to elucidate the transport of other fat-soluble vitamins. ABCA1, ABCG1, ABCG4, and ABCG5/ABCG8 transport cholesterol. In this study, neither ABCA1 nor ABCG1 appeared to transport vitamin K_3_, while ABCG5/ABCG8 did. This suggests that the substrate-binding pocket of ABCG5/ABCG8 differs from that of other cholesterol transporters. Vitamin K_3_ is reportedly a substrate of ABCG2 [17]. ABCG5, ABCG8, and ABCG2 belong to the G subgroup of the ABC transporter superfamily. Thus, the substrate-binding pockets of the proteins may be similar. It is unclear whether ABCG2 transports vitamin K_1_. If vitamin K1 is also a substrate of ABCG2, then ABCG2, in addition to ABCG5/ABCG8 and NPC1L1, may be involved in the regulation of vitamin K absorption. Furthermore, ABCB1 is involved in vitamin K_1_ transport [24]. Thus, multiple ABC transporters may be involved in regulating vitamin K absorption.

Vitamin K plays an important role in blood coagulation, because it is an essential cofactor of enzymes. Coadministration of ezetimibe, an NPC1L1 inhibitor, and warfarin, a vitamin K antagonist, greatly reduces vitamin K levels, which increases the risk of bleeding in extension of prothrombin time [12]. The expression and activity of ABCG5/ABCG8 may affect blood coagulation when warfarin is administered. Furthermore, cholesterol may affect vitamin K import via NPC1L1, because both cholesterol and vitamin K are substrates of NPC1L1 and may compete with each other. Thus, cholesterol and plant sterols may affect vitamin K transport by ABCG5/ABCG8, because they appear to be substrates of ABCG5/ABCG8. However, it is difficult to estimate the effect of sterols on vitamin K transport by ABCG5/ABCG8, because sterols affect membrane integrity and ABCG5/ABCG8 indirectly, as well as the solubility of vitamin K in the intestine and bile duct.

In summary, we showed that ABCG5 and ABCG8 are involved in vitamin K transport. This will shed light on the kinetics of vitamin K transport via transporters. Moreover, our results suggest that vitamin K absorption and excretion are regulated by NPC1L1 and ABCG5/ABCG8.

## Figures and Tables

**Figure 1 nutrients-15-00998-f001:**
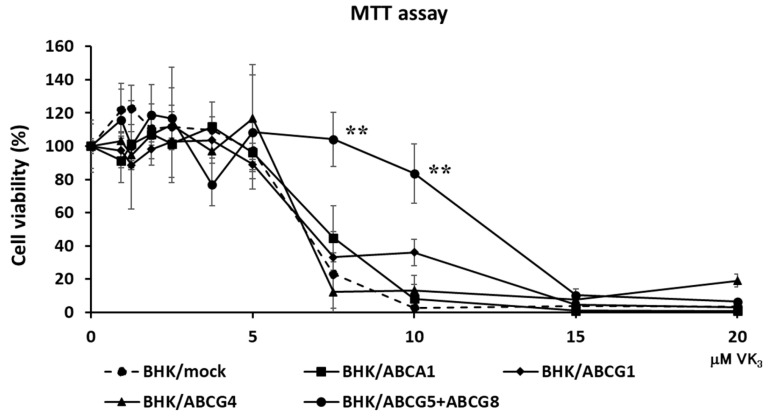
Cell viability against vitamin K_3_. BHK/mock, BHK/ABCA1, BHK/ABCG1, BHK/ABCG4, and BHK/ABCG5 + ABCG8 cells were incubated with indicated concentrations of vitamin K_3_ for 24 h. Vitamin K_3_ cytotoxicity was estimated using the MTT assay. The measurements were performed in triplicate and the average values of percentage cell viability ± SD are displayed. ** *p* < 0.01, compared to BHK/mock cells.

**Figure 2 nutrients-15-00998-f002:**
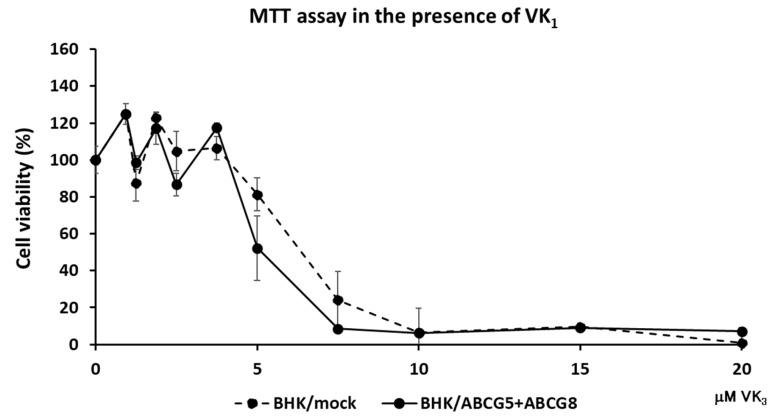
Cell viability against vitamin K_3_ in the presence of vitamin K_1_. BHK/mock and BHK/ABCG5 + ABCG8 cells were incubated with indicated concentrations of vitamin K_3_ in the presence of 100 μM vitamin K_1_ for 24 h. Vitamin K_3_ cytotoxicity was estimated using the MTT assay.

**Figure 3 nutrients-15-00998-f003:**
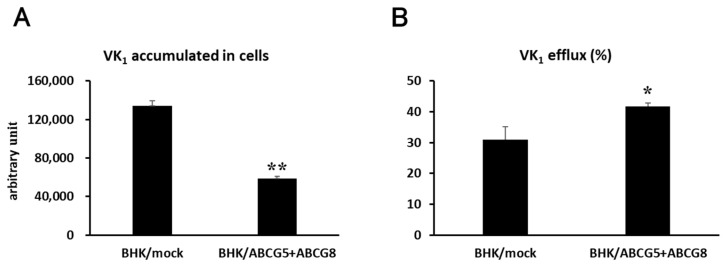
Vitamin K_1_ transport. BHK/mock and BHK/ABCG5 + ABCG8 cells were incubated with 100 μM vitamin K_1_ for 24 h. After washing, the cells were incubated for 4 h. (**A**) Vitamin K_1_ content in medium and cells was measured using HPLC. (**B**) Vitamin K_1_ efflux was calculated as vitamin K_1_ content in medium/(vitamin K_1_ content in cells + medium) × 100. * *p* < 0.05 and ** *p* < 0.01, compared to BHK/mock cells.

**Figure 4 nutrients-15-00998-f004:**
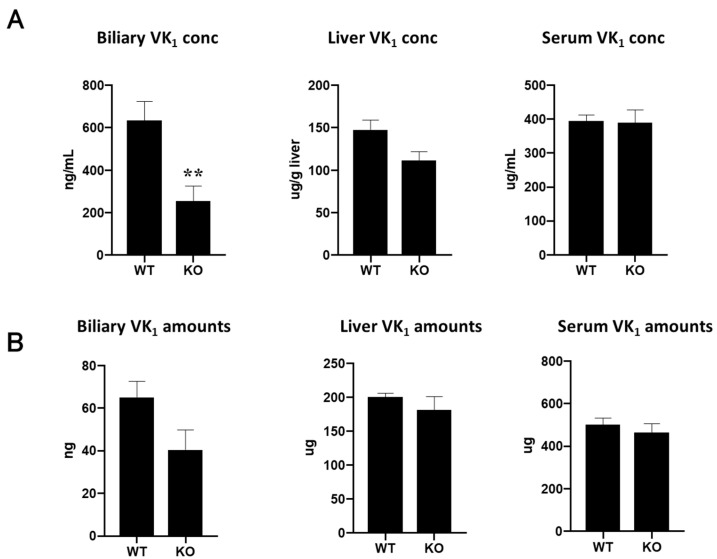
Vitamin K_1_ levels in mice bile, liver, and serum. (**A**) Vitamin K_1_ concentrations in bile, liver, and serum of wild-type (WT) and Abcg5/Abcg8-deficient (KO) mice were measured using UPLC analysis. (**B**) Vitamin K_1_ amounts in bile, liver, and serum were calculated by multiplying the concentration and volume or weight of each organ. The total serum volume was assumed to be 4% of the body weight. ** *p* < 0.01, compared to wild-type mice.

**Figure 5 nutrients-15-00998-f005:**
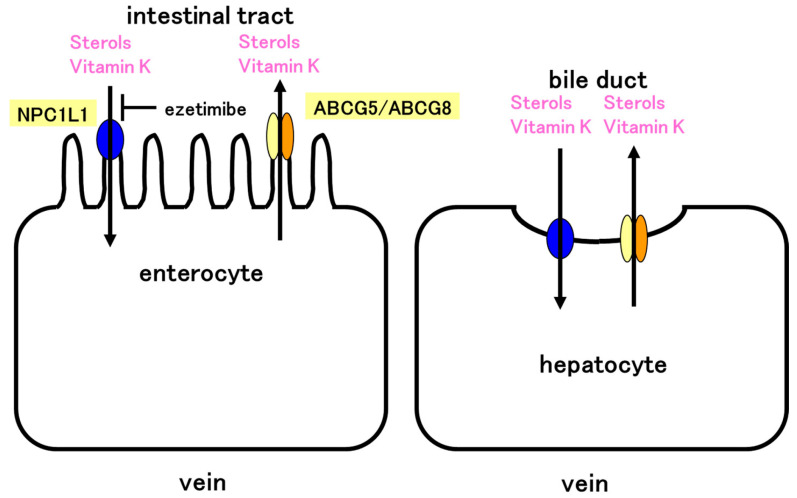
Model of vitamin K absorption in the intestine and excretion in the liver. NPC1L1 absorbs both sterols and vitamin K. The absorption is inhibited by ezetimibe. ABCG5/ABCG8 effluxes sterols and vitamin K. NPC1L1 and ABCG5/ABCG8 control absorption of vitamin K in the intestine. Similarly, NPC1L1 and ABCG5/ABCG8 control vitamin K excretion in the liver. ABCG5/ABCG8 efflux vitamin K and NPC1L1 would reabsorb vitamin K in the liver. NPC1L1 and ABCG5/ABCG8 may cooperatively regulate vitamin K levels in the body.

## Data Availability

The authors confirm that the datasets analyzed during the study are available from the corresponding author upon reasonable request.

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
