# Peer review of "ABCG5 and ABCG8 Are Involved in Vitamin K Transport"

_nutrients, 2023, doi:10.3390/nu15040998_

Round 1
Reviewer 1 Report
This is a very interesting and well written manuscript illustrating the role of ABCG5/ABCG8 transports of vitamin K. However, some points deserves to be improved. In particular:
Lines 284-289: It deserves to be highlithed that Vitamin K3 showed an important anti-cancer activity but resistance to this compound has also been reported in cancer (see PMID: 35901941). This is an important point to add since it can further highlight the interesting results found by the authors
Lines 28-33: references must be reported
2.5. Immunoblotting: Antibodies dilution and product codes must be added
To my opinion the supplementary Figure is not necessary and should be insert in the main manuscript
Author Response
Thank you for your valuable suggestions.
We added sentences concerning about anti-cancer activity of vitamin K3 to lines 314-317. “
We added references to lines 28-33 (lines 29-35 in our revised manuscript).
In 2.5 Immunoblotting section, we described antibodies dilution rates. We added product codes of antibodies in 2.1 Materials section.
We think that supplementary Figure is necessary to show that ABCG5/ABCG8 and other ABC transporters are functional. Because the data is not a topic of vitamin transport but cholesterol transport, we think that the figure is not appropriate as a main figure.
The comments were all very helpful in improving our manuscript, which we now hope is suitable for publication in Nutrients.
Reviewer 2 Report
See two comments in the PDF. Arteriosclerosis is shortly mentioned in the beginning - then not evolved further. Both vitamin K2 and effects on Matrix-Gla-Protein (MGP) and Growth arrest protein 6 (Gas6) could be mentioned - see reviews from Schurgers L et al. Transport of fat soluble vitamins from intestines to blood (micell production/transport) into liver and out again to different tissues is very complex and affected by genetics and nutrition and age and sex (genetics: eg ApoE). Recently total-cholesterol and levels of HDL and LDL and lowering these have been challenged for human health. Comments? The methodology used is very complex - has the the tests been used in humans? Have there been any studies with this methodology and vitamin k2 (menaquine 7?) administration. Last, rodents eat faeces (coprophagy) - so to control vitamin k intake you need to collect faeces with bags on the rear parts of the rats/mice - Comment?
